# Exploring the Promise of Second-Line Chemotherapy in Biliary Tract Tumours: A Glimpse into Novel Treatment Approaches

**DOI:** 10.3390/cancers15235543

**Published:** 2023-11-23

**Authors:** Paula Villalba Cuesta, Mercedes Avedillo Ruidiaz, Eva Ruiz Hispán, Raquel Fuentes Mateos, Angela Lamarca

**Affiliations:** 1Department of Medical Oncology–OncoHealth Institute, Fundación Jiménez Díaz University Hospital, 28040 Madrid, Spain; paula.villalba@quironsalud.es (P.V.C.); mercedes.avedillo@quironsalud.es (M.A.R.); eva.ruizh@quironsalud.es (E.R.H.); raquel.fmateos@quironsalud.es (R.F.M.); 2Department of Medical Oncology, The Christie NHS Foundation Trust, Manchester M20 4BX, UK

**Keywords:** advanced biliary cancer, cholangiocarcinoma, chemotherapy, second line, review

## Abstract

**Simple Summary:**

Biliary tract tumours are often diagnosed at an advanced stage and have relatively few treatment options with a poor prognosis. After first line therapy only a small fraction of patients receive further treatment, several clinical trials have been published on this matter and there is now a more clear consensus on second line therapy. Despite the new data, there are multiple ongoing challenges to face. Our review aims to revise the current evidence available on second line therapy in advanced biliary tract tumours.

**Abstract:**

Biliary tract tumours, including bile duct, gallbladder, and ampulla of Vater malignancies, pose a rare but formidable oncologic challenge. Typically diagnosed at advanced stages, these tumours offer limited treatment options and dismal prognoses, with a five-year survival rate below 20%. First-line chemotherapy with gemcitabine-cisplatin has demonstrated only modest efficacy, leaving a pressing need for improved therapeutic strategies. This comprehensive review provides a detailed examination of the current landscape of second-line chemotherapy for biliary tract tumours. The pivotal ABC-06 trial established FOLFOX (5-fluorouracil, leucovorin, and oxaliplatin) as the standard second-line therapy, demonstrating improved overall survival compared to active symptom control alone. Conversely, the NIFTY trial introduced nal-IRI (nanoliposomal irinotecan) plus 5-FU/LV (5-fluorouracil and leucovorin) as an alternative option, demonstrating substantial gains in progression-free and overall survival. However, the posterior NALIRICC trial presented conflicting results, raising questions about the added benefit of nal-IRI. Challenges in delivering second-line chemotherapy include rapid patient performance deterioration post-first-line treatment and limited access to second-line therapy. Only a fraction of eligible patients receive second-line therapy, emphasising the need for more effective first-line therapies to maintain patient fitness. The role of monotherapy in the second-line setting remains uncertain, particularly in unfit patients, and the absence of biomarkers for tailored treatment underscores the need for ongoing research. While challenges persist, ongoing investigations offer hope for optimising second-line therapy for biliary tract tumours, promising improved outcomes for patients facing this disease. This review provides an overview of current facts and challenges when delivering second-line chemotherapy for advanced biliary tract tumours.

## 1. Introduction

Biliary tract tumours, encompassing cancers arising in the bile ducts, gallbladder, and ampulla of Vater, represent a formidable challenge in oncology. Cholangiocarcinoma and ampullary carcinomas are more common in men, while gallbladder tumours are more frequent in women. They usually arise in patients aged between 50 and 70 years, and most tumours (>90%) are adenocarcinomas [1]. These malignancies are very uncommon, making up less than 1% of all cancers globally; nevertheless, studies over the past 25 years indicate that incidence and mortality are increasing, largely due to the increase in intrahepatic cholangiocarcinoma [2]. They often present at advanced stages, limiting treatment options and resulting in poor prognosis, with an overall survival rate at 5 years below 20% [3,4]

While first-line chemotherapy has shown modest efficacy, disease progression and resistance to initial treatment remain a significant concern. In recent years, the emergence of second-line chemotherapy regimens has brought a ray of hope to these patients [5]. This article aims to provide a comprehensive understanding of the potential impact of second-line chemotherapy in the fight against biliary tract tumours.

The first-line treatment for advanced biliary tract cancer involves a combination of chemotherapy. The ABC-02 trial investigated the efficacy of gemcitabine plus cisplatin compared to gemcitabine alone in 410 patients with locally advanced or metastatic bile duct, gallbladder, or ampullary cancer [6]. This trial demonstrated a significant improvement in overall survival with the gemcitabine-cisplatin combination, with a median survival of 11.7 months compared to 8.1 months in the gemcitabine-alone group. The combination therapy also resulted in improved progression-free survival, overall response rates, and a higher disease control rate. Similar results were observed in a separate randomised trial conducted in Japan, which also showed a survival benefit with the combination of gemcitabine and cisplatin compared to gemcitabine alone (11.2 months versus 7.7 months) [7]. These results established gemcitabine-cisplatin as the standard first-line treatment for advanced biliary tract cancer.

Improving the outcomes achieved with cisplatin and gemcitabine in the first-line setting has been challenging. Two clinical trials failed to show the benefit of triple chemotherapy compared to cisplatin and gemcitabine in this setting. The first study [8] explored the role of FOLFIRINOX in this setting, with no improved outcome over doublet therapy. The second study [9] did not show benefit from the addition of nab-paclitaxel to cisplatin and gemcitabine over cisplatin and gemcitabine alone. Thus, cisplatin and gemcitabine remain the backbone chemotherapy of choice in advanced biliary tract cancer.

The phase 3 TOPAZ-1 study investigated the combination of gemcitabine and cisplatin plus durvalumab in a double-blind, placebo-controlled trial involving 685 patients with previously untreated, unresectable locally advanced or metastatic biliary tract cancer [10]. After a median follow-up of 17 months, durvalumab plus chemotherapy demonstrated improved overall survival (HR 0.8, 95% CI, *p* = 0.021; median 12.8 months versus 11.5 months), progression-free survival (HR 0.75, 95% CI, *p* = 0.001; median 7.2 months versus 5.7 months), and objective response rate (27% versus 19%) compared to placebo plus chemotherapy. Based on these positive results, the US Food and Drug Administration (FDA) has approved durvalumab, in combination with gemcitabine and cisplatin, for the first-line treatment of adult patients with locally advanced or metastatic biliary tract cancers. Furthermore, in April 2023, the results of the Keynote 966 study were published [11]. Keynote-966 was a randomised, double-blind, placebo-controlled, phase 3 trial in which a total of 1069 patients were randomised to receive pembrolizumab plus gemcitabine and cisplatin or placebo plus gemcitabine and cisplatin. The primary endpoint was overall survival; secondary endpoints were progression-free survival, objective response rate, and duration of response. The median duration of follow-up was 25.6 months. The study showed a benefit in favour of the pembrolizumab arm in terms of overall survival (HR 0.83, 95% CI, *p* = 0.0034); median overall survival for the pembrolizumab group was 12.7 months versus 10.9 months in the placebo group. In contrast, the pre-specified statistical threshold for positivity was not met for progression-free survival (HR 0.86, 95% CI, *p* = 0.023; 6.5 months vs. 5.6 months) and objective response rates were similar in both arms (29% vs. 29%).

Based on this data, current guidelines [12,13] have adopted the combination of an immune checkpoint inhibitor (in the form of durvalumab, since pembrolizumab data were not available at the time) with cisplatin and gemcitabine as the new standard of care in the first-line setting. However, much improvement is still to be made seeing as medication resistance is a principal problem and there are inconsistent response rates among patients.

In the advanced setting, it is also important to identify patients who are potential candidates for targeted therapies seeing as they have emerged as potential treatment options and are a current area of interest in this field [14]. It is recommended to conduct targeted tests during or after first-line treatment in order to identify specific molecular abnormalities such as tumours with isocitrate dehydrogenase-1 (IDH-1) mutations, fibroblast growth factor receptor-2 (FGFR-2) fusions, human epidermal growth factor-2 (HER-2) amplifications, B-Raf proto-oncogene serine/threonine kinase (BRAF) V600E mutations, neurotrophic tyrosine receptor kinase (NTRK) fusions, MDM2 amplifications, and/or microsatellite instability. It is expected that around 40% of patients may have some form of targetable alterations [15], and the latest ESMO guidelines [13] would recommend that these options be explored after progression to first-line therapy if a targetable alteration has been identified.

Unfortunately, around 60% of patients will have no targetable alterations identified, and their treatment options will therefore be limited to second-line chemotherapy alternatives, this being the focus of our review.

## 2. Second-Line Chemotherapy for BTC: Where Are We Coming From?

Evidence on chemotherapy options in the second-line setting is scarce. Less than a decade ago there was no standard treatment for second-line therapy. A systematic review published in 2014 analysed data from 25 studies that included a total of 761 patients [16]. The mean overall survival was 7.2 months (95% CI 6.2–8.2) from commencement of second-line chemotherapy treatment, while in the ABC-02 trial, the overall survival after progression to first-line cisplatin and gemcitabine was around 4 months; however, this may be due to selection bias, as patients who are eligible and receive second-line chemotherapy need to be fit enough with a good performance status, and therefore they have a better prognosis; this has been shown in multiple retrospective and phase II trials [17,18,19].

The mean progression-free survival was 3.2 months (95% CI 2.7–3.7), response rate 7.7% (95% CI 4.6–10.9), and disease control rate 49.5% (95% CI 41.4–57.7). The systematic review concluded that while there was solid scientific support (level A evidence) for utilising initial chemotherapy in the treatment of advanced biliary cancer, there was comparatively scarce and less definitive evidence (level C evidence) when suggesting a second-line chemotherapy approach for these patients.

With such scarce data available, clinicians have tended to offer second-line chemotherapy based on scarce scientific grounds. In addition, only around 15–25% of patients were considered fit enough [16]. In fact, the prior version of the ESMO guidelines (2016) [20] did not consider second-line chemotherapy as a standard treatment option and patients were, on many occasions, offered the best supportive care only after progression to first-line chemotherapy.

However, in recent years, three clinical trials exploring second-line chemotherapy options have been published [21,22,23] (Table 1). There is a limited number of studies that directly compare different chemotherapy treatments for advanced biliary tract tumours. Additionally, there is no established consensus on which patients should receive second-line therapy or which treatment regimen is the most effective. Data of the most important studies will be discussed in the coming sections below.

## 3. ABC-06 Trial

The ABC-06 trial was the first prospective phase 3 randomised trial evaluating the advantage obtained from the use of second-line FOLFOX (5-fluorouracil, leucovorin, and oxaliplatin) treatment in advanced biliary tract cancer, including cholangiocarcinoma and gallbladder or ampullary carcinoma [21]. It was a phase 3, open-label, multicentre, randomised trial that revealed an improvement in overall survival (the primary outcome) when using FOLFOX compared to ASC (active symptom control). The primary endpoint was overall survival in the intention-to-treat population.

From 27 March 2014 to 4 January 2018, a total of 290 patients were evaluated for eligibility; among them, 162 patients were randomly divided into two groups: one receiving ASC (n = 81), and the other receiving ASC plus FOLFOX treatment (n = 81). Around 90% of patients in both study groups had adenocarcinoma histology. The study concluded on 4 January 2019, when the last recruited patient completed follow-up. By the data cut-off point, the median follow-up duration was 21.7 months (interquartile range 17.2–30.8).

The median overall survival was 6.2 months (95% CI 5.4–7.6) in the ASC plus FOLFOX group, compared to 5.3 months (4.1–5.8) in the ASC-alone group. These differences were modest, but statistically significant (adjusted HR 0.69, 95% CI 0.50–0.97, *p* = 0.031) and clinically relevant, with survival rates at 6 and 12 months of 50.6% (39.3–60.9) and 25.9% (17.0–35.8), respectively, with FOLFOX. Survival rates with ASC alone were 35.5% (95% CI 25.2–46.0) at 6 months, and 11.4% (5.6–19.5) at 12 months.

At the time of data analysis, 78 (96%) of the 81 patients assigned to ASC plus FOLFOX had either experienced disease progression or died. The median progression-free survival was 4.0 months (95% CI 3.2–5.0) with FOLFOX (no data available for the control arm) while the progression-free survival rate was 66.7% (95% CI 55.3–75.8) at 3 months, 32.1% (22.3–42.3) at 6 months, and 8.6% (3.8–16.0) at 12 months.

Among the patients in the ASC plus FOLFOX group, disease control was observed in 27 (33%) out of 81 patients, including 23 (28%) with stable disease. The remaining 53 patients were classified as non-responders: 30 (37%) had progressive disease, and 23 (28%) died. Three patients were still alive with stable disease at the time of the last follow-up, with individual progression-free survival ranging between 13.3 and 18.4 months.

The median number of cycles of FOLFOX administered was five. Among the 75 patients, 46 (61%) required a reduction or omission of at least one component of FOLFOX chemotherapy in one or more cycles. Only 13 (16%) out of 81 patients completed all 12 cycles of FOLFOX. The primary reasons for discontinuing treatment early included radiological disease progression (24 patients), clinical disease progression (13 patients), and intolerable toxicity (10 patients).

Grade 3–5 adverse events were documented in 56 (69%) out of 81 patients in the ASC plus FOLFOX group and in 42 (52%) out of 81 patients in the ASC-alone group. In the ASC plus FOLFOX group, three deaths related to chemotherapy were reported. The most commonly reported grade 3–5 chemotherapy-related adverse events were neutropenia (12%), fatigue (11%), and infection (10%).

In terms of quality of life, a posterior analysis of the results from the quality of life (QoL) and value of health (V-He) questionnaires [24] revealed that incorporating FOLFOX alongside ASC did not seem to result in a decline in the quality of life parameters that were measured. Conversely, patients in the ASC-alone group appeared to undergo a worsening of EQ-5D utility values and the majority of the QLQ-30 scales. There was an observed deterioration in symptoms such as nausea and pain, while these symptoms remained relatively unchanged in the ASC plus FOLFOX group.

Additionally, 21 (13%) out of 162 patients received subsequent systemic anticancer therapy after the trial: 5 (3%) in the form of phase 1 trials and 16 (10%) through the use of chemotherapy agents.

This trial established FOLFOX as the standard second-line therapy in advanced biliary tract cancer.

## 4. NIFTY

The NIFTY trial was a phase 2b, open-label, randomised trial conducted in five major medical centres in South Korea between 5 September 2018 and 31 December 2021 [22]. The primary endpoint was blinded independent central review (BICR)-assessed progression-free survival.

Patients who had progressed to gemcitabine plus cisplatin were randomly assigned at a 1:1 ratio to receive either nal-IRI (nanoliposomal irinotecan) plus 5-FU/LV (5-fluorouracil and leucovorin) compared with 5-FU/LV alone. During the time of the study, a total of 193 patients underwent screening to determine eligibility, of which 178 were randomly selected and divided into two groups: one group received nal-IRI plus 5-FU/LV (n = 88), and the other group received only 5-FU/LV (n = 90).

As of the data cut-off on 1 September 2020, the median follow-up period for patients in the complete analysis set was 11.8 months, with an interquartile range of 7.7 to 18.7 months. Out of the 88 patients in the nal-IRI plus 5-FU/LV group, 64 patients (73%) had died. Similarly, out of the 86 patients in the 5-FU/LV-alone group, 74 patients (86%) had died.

The median progression-free survival, as assessed by the BICR, was significantly longer in the group receiving nal-IRI plus 5-FU/LV (7.1 months, 95% CI 3.6–8.8) compared to the group receiving 5-FU/LV alone (1.4 months, 95% CI 1.2–1.5). The HR was 0.56 (95% CI 0.39–0.81), indicating a significant difference in favour of the nal-IRI plus 5-FU/LV group. This result was statistically significant (*p* = 0.0019). Interestingly, the updated analysis of the trial was presented in ESMO 2022 and included a longer follow-up period of 1.3 years [25]. A reperformed masked independent central review (MICR) was conducted, involving three newly invited radiologists. The median progression-free survival assessed by MICR was found to be 4.2 months (95% CI 2.8–5.3) for the group treated with nal-IRI plus 5-FU/LV. In contrast, the 5-FU/LV-alone group had a median progression-free survival of 1.7 months (95% CI 1.4–2.6). The HR was 0.61 (95% CI 0.44–0.86) and the *p*-value was 0.004, indicating a statistically significant improvement in progression-free survival for the nal-IRI plus 5-FU/LV group compared to the 5-FU/LV-alone group. These findings differ significantly from the previous report, and explanations for such differences remain unclear.

The median number of treatment cycles administered was six (interquartile range 3–12) for the nal-IRI plus 5-FU/LV group and three (interquartile range 3–6) for the 5-FU/LV-alone group. Adverse events led to the discontinuation of the study treatment in six patients in the nal-IRI plus 5-FU/LV and three patients in the 5-FU/LV-alone group. Dose modifications were necessary for 70 patients (80%) in the nal-IRI plus 5-FU/LV group and for 26 patients (30%) in the 5-FU/LV-alone group.

The most common grade 3–4 adverse events were neutropenia and fatigue. In total, 24% of patients in the nal-IRI plus 5-FU/LV group experienced neutropenia versus 1% in the 5-FU/LV-alone group. Fatigue was reported by 13% of patients in the nal-IRI plus 5-FU/LV group, compared to 3% in the 5-FU/LV-alone group. Serious adverse events were observed in 37 patients (42%) receiving nal-IRI plus 5-FU/LV. Among them, six patients (7%) experienced serious adverse events that were related to the treatment, including grade 4 pancytopenia, grade 3 febrile neutropenia, grade 3 diarrhoea, grade 3 fatigue, and grade 3 acute kidney injury. In the 5-FU/LV-alone group, serious adverse events were reported in 21 patients (24%), with one patient (1%) experiencing a treatment-related serious adverse event, specifically grade 3 colitis.

In terms of overall survival, the primary analysis reported a median overall survival of 8.6 months in the nal-IRI plus 5-FU/LV group versus 5.5 months in the 5-FU/LV-alone group (HR 0.68, 95% CI 0.48–0.98, *p* = 0.035) [22]. When data were updated in ESMO 2022 [25], the median overall survival was 8.6 months in the nal-IRI plus 5-FU/LV group versus 5.3 months in the 5-FU/LV-alone group (HR 0.68, 95% CI 0.48–0.95, *p* = 0.02).

Based on these results, the combination of nal-IRI plus 5-FU/LV was considered a potential alternative to FOLFOX in the second-line therapy for biliary tract tumours after progression to cisplatin and gemcitabine.

## 5. NALIRICC

The NALIRICC trial was a phase 2, prospective, randomised, two-sided trial [23]. This German study recruited eligible participants who were individuals over the age of 18, with an Eastern Cooperative Oncology Group performance status of 0 or 1, histologically confirmed metastatic biliary tract cancer, and progression after receiving first-line gemcitabine-based therapy.

This study aimed to assess the effectiveness of nal-IRI (nanoliposomal irinotecan) plus 5-FU/LV (5-fluorouracil and leucovorin) in comparison to 5-FU/LV (5-fluorouracil and leucovorin) alone in the second-line treatment of advanced biliary tract cancer.

The trial involved two treatment arms: Arm A received nal-IRI (80 mg/m^2^) in combination with 5-FU (2400 mg/m^2^)/LV (400 mg/m^2^) every two weeks, while Arm B received 5-FU/LV every two weeks. The primary endpoint of the study was progression-free survival, and secondary endpoints included overall survival, overall response rates, safety, and quality of life (measured by EORTC QLQ C30).

A total of 100 patients were randomly assigned in 17 centres, with 49 patients receiving nal-IRI plus 5-FU/LV treatment and 51 patients receiving 5-FU/LV-alone treatment. Among these patients, 64 had intrahepatic cancer, 19 had extrahepatic cancer, and 17 had gallbladder cancer. The characteristics of the patients were well balanced between the two treatment arms.

The NALIRICC did not meet its primary endpoint of progression-free survival; median progression-free survival in the nal-IRI plus 5-FU/LV group was 2.64 months versus 2.3 months in the 5-FU/LV-alone group (HR 0.867, 95% CI 0.559–1.345). Furthermore, the addition of nal-IRI to 5-FU/LV did not improve overall survival compared to 5-FU/LV alone: the median overall survival was 6.9 months versus 8.21 months, respectively (HR 1.082, 95% CI 0.681–1.720).

The most common grade 3 adverse events observed in the nal-IRI plus 5-FU/LV group were decreased neutrophil counts (16.6%), diarrhoea (14.6%), and nausea (8.3%). The combination of nal-IRI and 5-FU/LV was associated with a higher toxicity and a higher frequency of dose reductions. Quality of life (QoL) was found to be similar between both treatment arms.

Based on the results from the NALIRICC trial data, the option of 5-FU/LV alone was suggested by authors as an option [23], doubting whether the addition of nal-IRI was really of much value and therefore questioning the results from the NIFTY study.

## 6. Study Comparison

The ABC-06 trial included 162 patients, while the NIFTY trial had 174 patients, making them relatively large studies. Nonetheless, comparing the efficacy outcomes between the NIFTY and ABC-06 trials presents challenges due to discrepancies in the frequency of disease evaluation (occurring every 6 weeks versus every 3 months), the requirement for a measurable lesion (mandatory versus not necessary), and the disease status of participants. Despite the differences, both FOLFOX and nal-IRI plus FU/LV demonstrated comparable efficacy in second-line treatment in a biomarker-unselected population, making them potential options that could be sequentially used.

However, the posterior German phase 2 NALIRICC trial, which had a similar design to the NIFTY trial, did not show any improvement in survival outcomes when nal-IRI was added to 5-FU/LV, despite an improvement in objective response rate (14.3% vs. 3.9%). The differences in results between these trials might be attributed to variations in sample size (174 vs. 100 patients) and the proportion of patients with intrahepatic cholangiocarcinoma (64% vs. 43%). Furthermore, differences in ethnicity could also play a role, as the NIFTY study was conducted in South Korea, while the NALIRICC study was conducted in Germany. Given these compelling results, nal-IRI plus 5-FU/LV could potentially be regarded as a standard-of-care option for second-line regimens.

## 7. Challenges Encountered When Delivering Second-Line Chemotherapy in BTC

Delivering chemotherapy in the second-line scenario in biliary tract tumours remains challenging (Figure 1). Most importantly, results are modest and honest, and realistic discussions are of crucial importance when offering such an option to patients, to allow informed decision making.

Following the progression to first-line chemotherapy, patients with advanced biliary tract cancer frequently encounter a swift deterioration in their performance status, limiting their treatment options [26]. A very limited number of patients receive second-line therapy. A retrospective series analysing the usage of second-line chemotherapy following the ABC-06 clinical trial in which FOLFOX was established as the standard treatment in the second-line setting observed that only 26.62% of patients received this line of treatment [21,27]. This makes recruitment into second-line clinical trials challenging and highlights the importance of better first-line therapies that may allow patients to get fitter into second-line scenarios.

In addition, combination chemotherapy has been compared to no chemotherapy (ABC-06) or to single 5-FU/LV (NIFTY and NALIRICC); nevertheless, there are not many studies comparing two combination strategies. A small phase II study [28] evaluated whether FOLFIRI was superior to FOLFOX in the second-line treatment of advanced biliary tract cancer. One hundred and twenty patients were initially enrolled, with 114 receiving treatment (57 in the FOLFOX group and 57 in the FOLFIRI group). At a median follow-up of 10.7 months, the 6-month overall survival rate was 58.1% for FOLFOX and 46.0% for FOLFIRI. Median progression-free survival was 2.8 months for FOLFOX (95% CI, 2.3–3.3) and 2.1 months for FOLFIRI (95% CI, 1.3–2.9), *p* = 0.682. Median overall survival was 6.6 months for FOLFOX (95% CI, 5.6–7.6) and 5.9 months for FOLFIRI (95% CI, 4.3–7.5), *p* = 0.887. In conclusion, the trial found that FOLFIRI was tolerable but not superior to FOLFOX.

Thus, there is no established consensus on which chemotherapy options patients should receive in the second-line scenario or which treatment regimen is the most effective. Since the ABC-06 study was the only randomised study, and taking into account the contradictory findings from NIFTY and NALIRIC, FOLFOX is currently preferred by some guidelines (i.e., ESMO 2023). However, irinotecan-based therapy is considered a valid third-line option, or even an alternative to FOLFOX if the latter is contraindicated (i.e., neuropathy).

The role of monotherapy in the second-line setting remains unclear. Outcomes achieved with 5-FU/LV alone are actually similar to the control arm with ASC and no chemotherapy in the ABC-06 study. Thus, we could argue that for unfit patients, symptom management may be the best option, with a lower risk of complications and similar outcomes, acknowledging, of course, limitations for inter-study comparisons.

Finally, biomarkers for tailored chemotherapy strategies are missing. DNA damage repair (DDR) has been suggested as a potential marker of response to platinum, but results in biliary tract tumours remain unclear. The subgroup analysis in ABC-06 showed that the benefit from FOLFOX remained regardless of whether patients had platinum-resistant or platinum-sensitive disease [21]. In addition, recently presented data from the translational research associated with the ABC-06 study concluded that DDR did not seem to correlate with response to platinum therapy in biliary tract tumours and that it could, indeed, be a marker of poorer survival [29]. The study’s objectives were to evaluate the frequency of somatic mutations in DDR genes in advanced biliary tract cancer to assess whether these mutations could predict patient responses to platinum-based chemotherapy, retrospectively using first-line treatment data, and prospectively analysing second-line FOLFOX therapy outcomes. The study also sought to determine if DDR gene mutations could act as prognostic indicators for overall survival in advanced biliary tract cancer patients. Pathogenic mutations in DDR genes were observed in 37.29% of patients; however, the presence of these mutations did not significantly influence progression-free survival with either first-line cisplatin-gemcitabine therapy or second-line FOLFOX treatment. Patients with DDR-altered genes exhibited a notably shorter median overall survival (4.59 months) in comparison to those without these mutations (7.23 months), HR 2.63 (95% CI 1.48–4.67), *p* = 0.001.

On top of this, mechanisms of resistance to first-line chemotherapy and how second-line agents can overcome these are poorly described [30,31].

## 8. Conclusions

The findings from the ABC-06 study indicate that FOLFOX chemotherapy can enhance overall survival in patients with favourable performance status who have advanced biliary tract cancer and have previously received cisplatin and gemcitabine treatment [21]. On the other hand, the NIFTY trial demonstrated that the inclusion of nal-IRI and 5-FU/LV resulted in a significant improvement in both progression-free survival and overall survival compared to 5-FU/LV alone [22]. Conversely, the NALIRCC trial did not achieve its primary objective, as the addition of nal-IRI to 5-FU/LV did not lead to improved progression-free survival or overall survival when compared to 5-FU/LV alone [23].

The phase 3 ABC-06 trial was the first to provide evidence supporting the benefits of second-line chemotherapy compared to active symptom control alone in terms of overall survival for patients with advanced biliary tract cancer who had previously undergone first-line treatment with gemcitabine plus cisplatin. This is also the only phase III study in this setting and remains the option of choice after cisplatin and gemcitabine if there are no contraindications for its administration. Irinotecan-based therapy remains an alternative and valid option for subsequent therapy. The role of monotherapy is somehow more debatable.

The emergence of novel therapies and combinations in recent years has brought renewed hope to the treatment landscape of this disease. However, challenges still exist, and collaborative efforts will be essential in advancing our understanding of advanced hepatobiliary tumours and optimising second-line therapy to improve patient outcomes in this challenging disease setting. Additional trials involving a broader patient population are necessary to validate and further substantiate treatment in this setting.

## Figures and Tables

**Figure 1 cancers-15-05543-f001:**
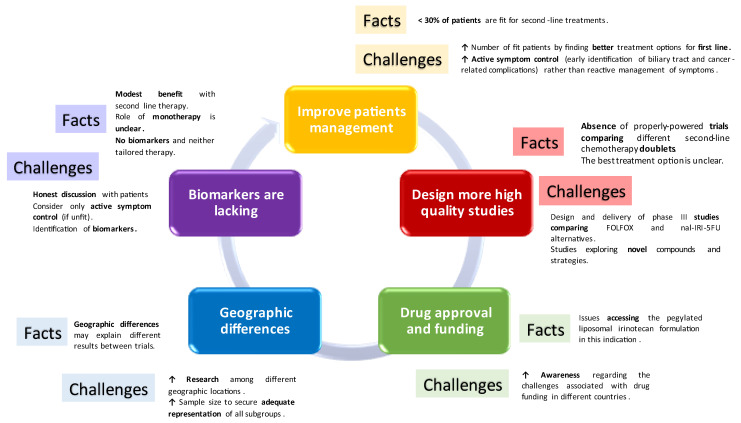
Challenges of second-line therapy in biliary tract tumours.

**Table 1 cancers-15-05543-t001:** Main clinical trials of second-line therapy in biliary tract tumours.

	ABC-06		NIFTY		NALIRICC	
	ASC Alone Group	ASC + FOLFOX Group	5-FU Alone Group	5-FU + Liposomal Irinotecan Group	5-FU Alone Group	5-FU + Liposomal Irinotecan Group
Type of study	Phase 3		Phase 2b		Phase 2	
N	81	81	86	88	51	49
Sex						
Male	37(46%)	43 (53%)	48 (56%)	51 (58%)	-	-
Female	44 (54%)	38 (47%)	38 (44%)	37 (42%)	-	-
Age (years) median	65 (59–72)	65 (59–72)	65 (37–80)	63 (38–84)	-	-
ECOG						
0	28 (35%)	25 (31%)	15 (17%)	23 (26%)	-	-
1	52 (64%)	55 (68%)	71 (83%)	65 (74%)	-	-
Unknown	1 (1%)	1 (1%)	-	-	-	-
Tumour site						
Intrahepatic	38 (47%)	34 (42%)	39 (45%)	35 (40%)	64 (64%)	
Extrahepatic	19 (23%)	26 (32%)	25 (29%)	22 (25%)	19 (19%)	
Gallbladder	17 (21%)	17 (21%)	22 (26%)	31 (35%)	17 (17%)	
Ampulla of Vater	7 (9%)	4 (5%)	-	-	-	
Disease stage						
Non-metastatic	15 (19%)	14 (17%)	0	0	-	-
Metastatic	66 (81%)	67 (83%)	86 (100%)	88 (100%)	-	-
Had previous surgery	38 (47%)	34 (42%)	29 (34%)	26 (30%)	-	-
Median follow-up	21.7 months (IQR 17.2–30.8)		11.8 months (IQR 7.7–18.7); extended follow-up of 1.3 years		5.9 months	
ORR	-	5%	5.8% (updated 3.5)	14.8% (updated 12.5)	3.9%	14.3%
Median PFS	-	4 months	1.4 months (updated 1.7)	7.1 months (updated 4.2)	2.3 months	2.76 months
Median OS	5.3 months	6.2 months	5.5 months (updated 5.3)	8.6 months (updated 8.6)	8.21 months	6.9 months
HR for OS (95% CI)	0.69 (0.50–0.97); *p* = 0.031		0.68 (0.48–0.98); *p* = 0.035; updated 0.68 (0.48–0.95), *p* = 0.02		-	-
HR for PFS (95% CI)	-	-	0.56 (0.39–0.81), *p* = 0.0019; updated 0.61 (0.44–0.86), *p* = 0.004		-	-
Grade 3–5 AEs	52%	69%	24%	42%	50%	70.8%
Patients with subsequent line of therapy	13%		11%	17%	-	-

ECOG: Eastern Cooperative Oncology Group. ORR: objective response rate. PFS: progression-free survival. OS: overall survival. HR: hazard ratio. AEs: adverse events. ASC: active symptom control. FOLFOX: 5-fluorouracil, leucovorin, and oxaliplatin. 5-FU: 5-fluorouracil.

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
