# Peer review of "Exploring the Promise of Second-Line Chemotherapy in Biliary Tract Tumours: A Glimpse into Novel Treatment Approaches"

_cancers, 2023, doi:10.3390/cancers15235543_

Round 1

Reviewer 1 Report

Comments and Suggestions for Authors

Title: Exploring the Promise of Second-Line Chemotherapy in Biliary Tract Tumours: A Glimpse into Novel Treatment Approaches

This paper describes an overview on current facts and challenges when delivering second-line chemotherapy for advanced biliary tract tumors.

This paper including relatively large number of references, but there are some questions and the author is requested to add the description according to comments as below.

Major points

a selective fibroblast growth factor receptor (FGFR) inhibitor in BTC

Pemigatinib is expected to contribute to second-line drug treatment after failure of standard therapies in biliary tract cancer.

The author should add a selective fibroblast growth factor receptor (FGFR) inhibitor in BTC for further understanding in advanced biliary tract tumours.

Author Response

Thank you very much for your valuable feedback. We greatly appreciate your input. Our review paper focuses on the second-line chemotherapy treatment of patients with advanced biliary tract tumours (BTC). In our introduction we have briefly mentioned the role of targeted therapies, however, seeing as this topic is so broad, we decided not to delve into the different treatments available. If we mention specific FGFR inhibitors, we would also have to mention the rest of targeted therapies on offer in the market at time of publication. We prefer to focus on chemotherapy strategies. 

Reviewer 2 Report

Comments and Suggestions for Authors

It would be better to describe whether ABC-06 trial, NIFTY trial, and  NALIRICC trial are scientific and explain in detail.

It would be better to describe which type of biliary tract tumors (for example, adenocarcinoma, adenosquamous carcinoma, neorendcorine carcinoma, etc.).

There are minor comments.

For example, lines 35, Keywords:  remove "Keywords:"

                     In Table 1, Ampulla -> Ampulla of Vater                               

                         Please indicate the abbreviation below Table 1.

Comments on the Quality of English Language

Please check Englsih grammar.

Author Response

Thank you very much for your valuable feedback. We greatly appreciate your input.

Regarding your different suggestions/comments:

  • "It would be better to describe whether ABC-06 trial, NIFTY trial, and  NALIRICC trial are scientific and explain in detail". We have added extra information about each of the clinical trials in the text.
  • "It would be better to describe which type of biliary tract tumors (for example, adenocarcinoma, adenosquamous carcinoma, neorendcorine carcinoma, etc.)". We have added this information in the section on the ABC-06 trial. This information is not available for the NALIRICC or NIFTY trials, therefore we are unable to add it.
  • "Lines 35, Keywords:  remove "Keywords". We have deleted this word.
  • "In Table 1, Ampulla -> Ampulla of Vater". We have changed this.
  • "Please indicate the abbreviation below Table 1". We have indicated the meaning of all the abbreviations below the table.

We have also reviewed the grammar.

We are open to any further suggestions you may have. 

Reviewer 3 Report

Comments and Suggestions for Authors

This review article, "Exploring the Promise of Second-Line Chemotherapy in Biliary Tract Tumours: A Glimpse into Novel Treatment Approaches," is summarized in this paper. The document provides background for the challenges and opportunities in treating biliary tract cancers as well as an overview of the main themes discussed in the research.

Comments:

2- In the introduction, the subject of biliary tract malignancies, which comprise cancers of the bile ducts, gallbladder, and ampulla of Vater, first Highlight the significance of biliary tract tumors as a difficult but relatively uncommon type of cancer. To highlight the clinical significance of these malignancies, provide statistics on their incidence and prevalence. Also, emphasize how the prognosis for biliary tract cancers is often poor, underscoring the demand for efficient therapies.

2- Explain that the current standard of care for these malignancies is first-line chemotherapy, frequently based on the gemcitabine/cisplatin regimen. Explain about the drawbacks and difficulties of first-line therapies, such as medication resistance and inconsistent response rates.

3- Please also discuss emerging treatment approaches, like immunotherapy, targeted therapeutics, and customized medicine. Describe the ways in which innovative drugs and therapies work to get beyond the drawbacks of conventional medicine. In addition, provide evidence from clinical trials or case studies demonstrating the effectiveness of these novel treatments.

4-Explain the mechanism of action of second-line chemotherapeutic drugs, paying special attention to their capacity to overcome resistance mechanisms created during first-line therapy.

5-Explain the mechanisms of resistance that can arise during second-line therapy and possible countermeasures.

Comments on the Quality of English Language

 Minor editing is needed

Author Response

Thank you very much for your valuable feedback. We greatly appreciate your input.

Regarding your different suggestions:

1. In the introduction, the subject of biliary tract malignancies, which comprise cancers of the bile ducts, gallbladder, and ampulla of Vater, first Highlight the significance of biliary tract tumors as a difficult but relatively uncommon type of cancer. To highlight the clinical significance of these malignancies, provide statistics on their incidence and prevalence. Also, emphasize how the prognosis for biliary tract cancers is often poor, underscoring the demand for efficient therapies.

In the first paragraph of the introduction section we have tried to highlight that BTC is an uncommon type of cancer with a poor prognosis.

2. Explain that the current standard of care for these malignancies is first-line chemotherapy, frequently based on the gemcitabine/cisplatin regimen. Explain about the drawbacks and difficulties of first-line therapies, such as medication resistance and inconsistent response rates.

The third, fourth, and fifth paragraphs of the introduction aim to address this point. We have added changes.

3. Please also discuss emerging treatment approaches, like immunotherapy, targeted therapeutics, and customized medicine. Describe the ways in which innovative drugs and therapies work to get beyond the drawbacks of conventional medicine. In addition, provide evidence from clinical trials or case studies demonstrating the effectiveness of these novel treatments.

We have modified the sixth paragraph of the introduction to clarify this point and further explain emerging treatment strategies.

4. Explain the mechanism of action of second-line chemotherapeutic drugs, paying special attention to their capacity to overcome resistance mechanisms created during first-line therapy.

5. Explain the mechanisms of resistance that can arise during second-line therapy and possible countermeasures.

We have added a paragraph at the end of the section “Challenges encountered when delivering second-line chemotherapy in BTC” in order to address point 4 and 5.

We are open to further suggestions.

Thank you very much in advance.